# A Pilot Study Assessing a Closed-Loop System for Goal-Directed Fluid Therapy in Abdominal Surgery Patients

**DOI:** 10.3390/jpm12091409

**Published:** 2022-08-30

**Authors:** Yann Gricourt, Camille Prin Derre, Christophe Demattei, Sébastien Bertran, Benjamin Louart, Laurent Muller, Natacha Simon, Jean-Yves Lefrant, Philippe Cuvillon, Samir Jaber, Claire Roger

**Affiliations:** 1IMAGINE, UR-UM 107, University of Montpellier, Division of Anaesthesia Critical Care, Pain and Emergency Medicine, Nîmes University Hospital, 30900 Nîmes, France; 2Laboratoire de Biostatistique, Epidémiologie Clinique, Santé Publique Innovation et Méthodologie (BESPIM), Pôle Pharmacie, Santé Publique, Nîmes University Hospital, 30900 Nîmes, France; 3Polyclinique Grand Sud, 350 Avenue Saint-André de Codols, 30000 Nîmes, France; 4Département d’Anesthésie Réanimation B St Eloi Hospital, 80 Avenue Augustin Fliche, 34295 Montpellier, France

**Keywords:** high-risk surgery, intraoperative fluid optimisation, closed-loop

## Abstract

Background: This prospective multicentre pilot study of patients scheduled for elective major abdominal surgery aimed to validate the fluid challenge (FC) proposed by the closed-loop (CL) system via anaesthesiologist assessment. Methods: This was a phase II trial consisting of two inclusion stages (SIMON method). Each FC (250 mL saline solution for 10 min) proposed by the CL was systematically validated by the anaesthesiologist who could either confirm or refuse the FC or give FC without the CL system. A ≥ 95% agreement between the CL and the anaesthesiologist was considered acceptable. Results: The study was interrupted after interim analysis of the first 19 patients (10 men, median age = 61 years, median body mass index = 26 kg/m^2^). The anaesthesiologists accepted 165/205 (80%) of fluid boluses proposed by the CL. Median cardiac index (CI) was 2.9 (interquartile: IQ (2.7; 3.4) L/min/m^2^) and the median coefficient of variation (CV) for CI was 13% (10; 17). Fifteen out of nineteen patients (79%) had a mean CI > 2.5 L/min/m^2^ or spent > 85% surgery time with pulse pressure variation < 13%. No adverse events related to the CL were reported. Conclusion: In this study of patients scheduled for elective major abdominal surgery, the agreement between CL and anaesthesiologist for giving fluid challenge was 80%, suggesting that CL cannot replace the physician but could help in decision making.

## 1. Background

In intermediate and high-risk surgery patients, intraoperative hemodynamic optimisation based on cardiac output (CO) and preload variables, such as stroke volume variations (SVV), reduces postoperative complications and length of stay (LOS) in hospital and saves costs [1,2,3,4,5,6,7,8,9]. Intraoperative goal-directed therapy (GDT) is therefore recommended as part of an enhanced recovery protocol [10,11,12]. However, GDT is still poorly applied in anaesthetic settings [13,14,15,16]. Even in studies aimed at assessing GDT, the use of GDT varies from 60 to 87.3% [14,17]. 

To improve compliance with GDT protocols, a closed-loop (CL) fluid administration and hemodynamic management system based on pulse pressure variation (PPV) and CO monitoring, initially called learning intravenous resuscitator, was first described by Rinehart et al. [18] in 2011. This system consisted of automatically titrating fluid administration (without human intervention) based on advanced hemodynamic variables such as SVV and PPV provided by continuous monitoring throughout the surgical procedure in order to maintain the patient in a preload-independent state and reach an optimal stroke volume value. The CL is able to perform repetitive tasks with a certain degree of autonomy, facilitating GDT in the operating room. This system was then initially validated for the optimization of CO and hemodynamic variables in experimental and pilot studies [19,20]. Interestingly, the variability of cardiac output was decreased during surgical procedures with this system and was shown to be as efficient as GDT managed by an anaesthesiologist [21].

However, further trials to assess the CL’s decision to administer fluid to the patient were made by the anaesthesiologist involved in patient management. This prospective phase II study was therefore aimed at assessing the CL system’s administration of fluid compared with the anaesthesiologist’s usual management, considered as the “gold standard” for fluid administration. 

## 2. Materials and Methods

This prospective study was performed at two tertiary university hospitals (Montpellier and Nîmes, France) and was approved by the local ethics committee of Nîmes (Comité de Protection des Personnes 2013.09.10) [22]. It was registered on ClinicalTrials.gov under the identification number NCT02138942. Written informed consent was obtained from patients prior to enrolment. 

All patients with an American Society of Anaesthesiologists (ASA) status ≤ 3 undergoing elective major abdominal surgery (laparotomy or laparoscopy) with an anticipated duration ≥ 2 h were eligible. All patients ≥18 and ≤ 75 years old requiring an arterial catheter for CO and pulse pressure variation (PPV) optimization were included. Patients were not included if they refused to participate or were under guardianship. Pregnant women and any patients with ASA status > 3, cardiac arrhythmia, or right ventricular failure were also excluded. 

### 2.1. Intraoperative Management

General anaesthesia was induced with propofol (1–2 mg/kg) and remifentanil (dose at the discretion of the anaesthesiologists in charge) and maintained with sevoflurane (0.8 to 1.2 targeted minimal alveolar concentration), remifentanil, and a neuromuscular blocking agent. After tracheal intubation, all patients were ventilated using a tidal volume of 8 to 9 mL/kg of ideal body weight, with a positive end expiratory pressure of 5 cm H_2_O and a respiratory rate adjusted to achieve an end-tidal CO_2_ of 32 to 35 cm H_2_O. Maintenance fluid was provided by 6 mL.kg^−1^.h^−1^ crystalloid infusion rate throughout surgery. Blood products were given to maintain a haemoglobin range of ≥7 g/dL. 

### 2.2. Closed-Loop System Description and Setup

The CL was under the supervision of the anaesthesiologist in charge of the patient.

The arterial line was connected to the hemodynamic Pulsion^®^ monitor (ProAQT Pulsion Medical Systems SE, Feldkirchen, Germany), itself connected to the CL with an USB-to-serial adapter (Figure 1).

The CL controlled an infusion pump (QCore^®^ Supphire Pump, Netanya, Israel) allowing fluid boluses to be administered to the patient through a venous line. The CL software was run on a Shuttle X50 touchscreen PC (Shuttle Computer Group, City of Industry, CA, USA) under Windows 7 (Microsoft Corp., Redmond, WA, USA). During the surgical procedure, the CL algorithm continuously tracked changes in different parameters: arterial pressure, heart rate (HR), CO, stroke volume (SV), and preload variables such as SVV or PPV to determine whether fluid administration would induce fluid responsiveness [19]. The CL algorithm uses a model based on a dataset of more than 400 patients (hemodynamic parameters before and after a 500 mL bolus of 6% hydroxyethyl starch) to predict the hemodynamic response to a 250 mL fluid bolus [19]. As the intervention progressed, the algorithm analysed the effects of its boluses and modified the patient predictions like an adaptive controller. The target was a 15% increase in CO in response to a fluid bolus of 500 mL. However, the CL always administers a maximum fluid bolus of 100 mL at a rate of 1000 mL.h^−1^ (corresponding to 100 mL in 6 min) with isotonic crystalloid (serum saline or lactate ringer). The algorithm was similar during laparoscopy and laparotomy procedures.

### 2.3. CL and Anaesthesiologist Agreement

As this study aimed to assess the agreement between the CL and the anaesthesiologist (like a safety assessment), the fluid boluses recommended by the CL were systematically checked before administration by the senior anaesthesiologist in charge of the patient. The anaesthesiologist could stop the fluid bolus administration if he considered it inappropriate, as defined by the following criteria:Invalid preload parameters,Poor quality of the pulse contour signal,No hemodynamic signs of hypovolemia,Any dysfunction of the device.

Agreement or disagreement between the CL and the anaesthesiologist were systematically recorded. 

### 2.4. Data Collection

Throughout the operative period, mean arterial pressure (MAP), heart rate (HR), intraoperative urine output, mean and optimal cardiac index (CI), stroke volume (SV), stroke volume index (SVI), and PPV were systematically recorded. The optimal CI was defined as the value of CI reached after a 250 mL fluid bolus leading to <15% CI increase. Fluid volumes (crystalloids, colloids) administered peri-operatively and red blood cell (RBC) transfusions were recorded. The patient’s arterial lactate level was measured at the end of surgery (normal range: (0.5–1.6) mmol/L). Postoperative complications, hospital LOS, and high dependency bed (HDB) LOS were also recorded. 

### 2.5. Study Outcomes

The primary outcome of this prospective study was the proportion of fluid boluses delivered by the CL and refused by the anaesthesiologist out of all the fluid boluses administered throughout surgery. An *a priori* compliance of ≥95% to be reached at the interim analysis was considered by the authors and independent expert committee as the primary endpoint for safety. 

The secondary outcomes were: -The proportion of fluid boluses administered by the anaesthesiologist in charge of the patient beyond the boluses recommended by CL,-The percentage of surgery time spent beyond the optimal CI and above 2.5 L/min/m^2^ with a PPV value < 13% (preload-independent state),-CI variability throughout the surgery defined by the CI’s coefficient of variation (CV).

### 2.6. Statistical Analysis 

This phase II trial had been designed with two inclusion stages [23]. To reach the second stage of inclusion, at least 17 of the 19 patients had to be considered as validated. In the second stage, more than 38 of the 42 patients had to be considered as validated for the CL system to be considered as clinically relevant. Assuming α < 0.05 and β < 0.1, we calculated a sample size of 42 patients. An interim analysis was planned after 19 patients with agreement to carry on if at least 17 patients had been validated.

Data were expressed as medians with interquartiles (IQ) or means ± standard deviation (SD) as appropriate. Continuous data were tested with the D’Agostino and Pearson normality test. Comparisons were made using the Mann–Whitney–Wilcoxon test and Fisher’s exact test for quantitative and qualitative data, respectively. A *p* < 0.05 was considered significant. Statistical analysis was performed by Graphpad Prism 5 for Windows. 

## 3. Results

Patient demographic data are shown in Table 1.

The independent expert committee decided to stop the study after the planned interim analysis (first 19 patients), as in 8/19 (42%) patients, less than 95% of the fluid challenges proposed by the CL were accepted by the anaesthesiologist (initial objective ≤ 2 patients).

Figure 2 and Table 2 show all the fluid challenges proposed by the CL for each patient with agreement or refusal by the anaesthesiologist. For 11 patients (58%), ≥95% of the fluid boluses proposed by the CL were accepted by the anaesthesiologist.

Overall, the CL proposed two hundred and five 100 mL fluid boluses (median = 9 (5; 17) boluses per patient), and the anaesthesiologist accepted one hundred and sixty-five (80%) of them. Among the 40 fluid boluses refused by the anaesthesiologist, 26 (65% of refused fluid, 13% of the overall fluid boluses) were considered unwarranted based on invalid preload parameters (arrhythmia occurring during surgery, poor pulse contour signal, or monitoring disturbed by surgery) and 14 (35% of refused fluid boluses, 7% of the overall fluid boluses) were considered unwarranted based on the hemodynamic data displayed. In five separate patients, the anaesthesiologist initiated one bolus outside the CL recommendation.

The secondary outcomes are shown in Table 1. Mean CI was >2.5 L/min/m^2^ in 15 (79%) patients, but only seven patients spent ≥ 85% of surgery time with a PPV < 13%. At the end of the surgical procedure, arterial lactate was ≤1.2 mmol/L in more than 75% patients. Postoperatively, two cases of septic shock and one surgical site infection were recorded with no adverse effect related to CL. 

## 4. Discussion

In the present study, 8/19 (42%) patients had less than 95% of fluid challenges proposed by the CL accepted by the anaesthesiologist (initial objective ≤ 2 patients), leading to the interruption of inclusions, as the initial hypothesis had been that 17 of the first 19 patients would have >95% compliance. However, the anaesthesiologist agreed to 80% of boluses proposed by the CL. Mean intraoperative CI was > 2.5 L.min^−1^.m^−2^ in 15/19 patients. At the end of the surgery, arterial lactate level was ≤1.2 mmol/L in 75% of patients. 

No previous studies assessing the CL system have been of similar design. Indeed, the initial development of the CL involved simulation and experimental studies [18,19,20]. Next, cohort studies showed that using the CL was associated with greater intraoperative hemodynamic stability with CI > 2.5 L/min/m^2^ and a preload-independent state [20,21]. Recently, various studies have reported that the CL has made it possible to increase the times with increased stroke volume and preload-independent state [24,25]. However, no differences in complication rates or length of hospital stay have been reported. A more complex computer-assisted closed loop (controlling fluid administration and vasopressor infusion) led to fewer episodes of intraoperative hypotension and a higher stroke volume index with no impact on postoperative complications [26].

The novelty of the present study resides in the fact that the anaesthesiologist in charge of the patient had to validate each fluid challenge proposed by the CL. This kind of study has never been previously performed. The study reported a disagreement between the CL and the anaesthesiologist for 8/19 (42%) patients, whereas over 80% of the fluid challenges proposed by the CL were accepted by the anaesthesiologist. These apparently “negative” findings might be explained as follows:

Disagreement between the CL and the anaesthesiologist was 38–47% in three of the first five patients, suggesting suboptimal expertise with CL even though the anaesthesiologist had followed training before initiating the study. This suboptimal expertise could probably explain why 7/19 patients spent more than 85% of surgery time with a PPV < 13%.

The objective of compliance > 95% was probably too stringent. This meant that no disagreement could exist in the first 20 fluid challenges. This objective was probably too ambitious when the physician’s ability to clinically predict fluid responsiveness is about 50% [27,28]. Interestingly, 80% of fluid challenges were not refused or stopped in the present study, indicating that the anaesthesiologist’s opinion was not challenged. However, it has been well-documented that cardiac output is rarely monitored in the operating room and that the most widely used indicators of volume expansion are clinical experience, blood pressure, or urine output [13,14]. 

One may question whether the anaesthesiologist involved in our study was sufficiently well-trained to implement goal-directed fluid therapy [21]. Indeed, only seven patients spent ≥ 85% of their surgery time with a PPV < 13%. However, the mean CI was >2.5 L/min/m^2^ in 79% of patients, and arterial lactate was normal in over 75% of patients. These findings probably mean that there was a gap between CL and the anaesthesiologist in the fluid administration strategy, with no impact on patient outcome. Interestingly, two studies assessing GDT in surgical procedures reported an implementation rate of 60 to 87.3%, which is similar to the agreement between the CL and the anaesthesiologist in our study [18,19].

In clinical practice, the present study shows that the CL system cannot replace an anaesthesiologist but should be considered as his (or her) intraoperative help for implementing goal-directed fluid therapy [29]. The 80% agreement reported in the present study would appear to recommend the use of CL for patients with no risk of fluid overload. The present findings tend to recommend a learning period of 5–10 patients before managing surgical procedures with this system. Indeed, the challenge is not to check each fluid challenge proposed by the CL but rather to stop the proposal when the conditions for validity are not applied, and the risk of fluid overload is too high for the anaesthesiologist. 

In conclusion, the present study showed that a >95% agreement between the CL system and the anaesthesiologist was an over-ambitious objective. Therefore, this system cannot replace the anaesthesiologist but instead help him (or her) for giving fluid challenge during the surgical procedure.

## Figures and Tables

**Figure 1 jpm-12-01409-f001:**
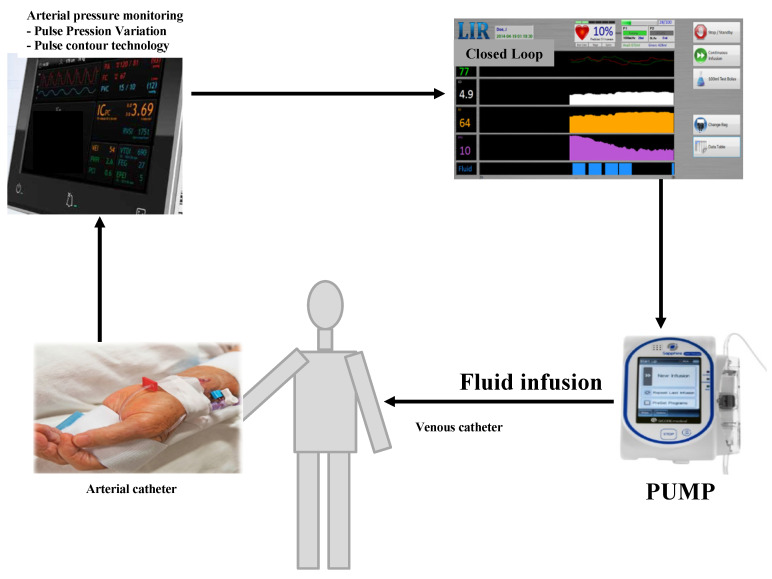
The closed-loop system.

**Figure 2 jpm-12-01409-f002:**
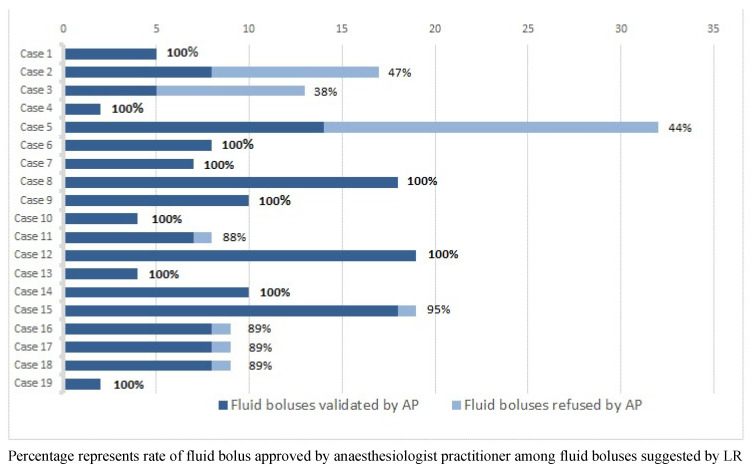
Agreement between the closed-loop system and the anaesthesiologist.

**Table 1 jpm-12-01409-t001:** Patient characteristics and secondary outcomes.

**BASELINE**	
Age (years)	61 (49–68)
Sex, No (%)	
Men	10 (53%)
Height (cm)	168 (162–176)
Weight (kg)	69 (62–80)
BMI (kg/m^2^)	26 (22–28)
**Comorbidities, N (%)**	
Hypertension	7 (37%)
Complicated diabetes	1 (5%)
Obesity (BMI > 30)	2 (10.5%)
**ASA, N (%)**	
ASA I	1 (5%)
ASA II	14 (74%)
ASA III	4 (21%)
ASA IV	0
**Surgical procedure**	
Type of procedure, N (%)	
Bowel resection	8 (42%)
Pancreaticoduodenectomy	5 (26%)
Adrenalectomy	2 (11%)
Splenopancreatectomy	2 (11%)
Others	2 (10%)
Coloscopy	10 (53%)
Laparoscopy	9 (47%)
Epidural analgesia	8 (42%)
**Biology**	
Baseline creatinine (µmol/L)	73 (59–81)
**Secondary Outcomes**	
Overall median CI L/min/m^2^	2.9 (2.7–3.4)
Patients with mean CI was >2.5 L/min/m^2^	15 (79%)
Patients who spent ≥85% of their surgery time with a PPV <13%	7/19 (37%)
Median CI coefficient of variation	13 % (10–17)
Median arterial lactate at the end of surgery (mmol/L)	1.1 (0.9–1.2)
Median serum creatinine on Day 1 (µmol/L)	62 (56–76)
Median length of hospital stay (days)	10 (5–14)
Median high dependency bed length of stay (days)	4 (0–7)
Median time to oral liquid (days)	2 (0.5–6)
Median time to solid intake (days)	3 (1–9.5)

(n = 19).

**Table 2 jpm-12-01409-t002:** Individual data for studied parameters per patient.

	HR (bpm)	MAP (mmHg)	CI (L.min^−1^.m^−2^)	SVI (mL.m^−2^)	CV of CI (%)	% Time with CI ≥ 2.5 L.min^−1^.m^−2^	%Time with PPV < 13%
Patient # 1	63 ± 4	72 ± 9	2.74 ± 0.25	43 ± 3	9	89	92
Patient # 2	84 ± 6	98 ± 12	3.84 ± 0.53	46 ± 5	14	98	40
Patient # 3	75 ± 10	108 ± 12	2.35 ± 0.41	31 ± 2	17	45	72
Patient # 4	55 ± 4	85 ± 17	3.64 ± 0.47	37 ± 3	13	100	93
Patient # 5	99 ± 6	68 ± 8	5.55 ± 0.79	56 ± 9	14	100	42
Patient # 6	68 ± 8	81 ± 10	1.92 ± 0.35	28 ± 3	18	0	68
Patient # 7	86 ± 9	60 ± 9	4.38 ± 0.55	51 ± 2	13	100	90
Patient # 8	63 ± 5	76 ± 11	2.89 ± 0.37	45 ± 4	13	92	10
Patient # 9	59 ± 5	72 ± 10	2.28 ± 0.24	38 ± 3	11	14	82
Patient # 10	75 ± 4	84 ± 7	3.07 ± 0.28	41 ± 2	9	100	94
Patient # 11	70 ± 4	80 ± 6	3.05 ± 0.25	44 ± 3	8	99	39
Patient # 12	63 ± 11	68 ± 11	2.26 ± 0.39	36 ± 3	17	32	46
Patient # 13	65 ± 5	69 ± 9	2.93 ± 0.48	45 ± 5	16	84	33
Patient # 14	60 ± 7	76 ± 20	2.89 ± 0.53	48 ± 9	18	76	89
Patient # 15	74 ± 9	73 ± 8	2.66 ± 0.34	36 ± 2	13	71	76
Patient # 16	59 ± 8	88 ± 11	2.73 ± 0.43	47 ± 4	16	75	53
Patient # 17	78 ± 7	78 ± 14	3.10 ± 0.32	40 ± 2	10	96	79
Patient # 18	71 ± 3	72 ± 9	3.41 ± 0.31	48 ± 4	9	100	98
Patient # 19	84 ± 15	97 ±9	2.84 ± 0.48	34 ± 2	17	80	98

#: patient number. HR, heart rate; MAP, mean arterial pressure; CI, cardiac index; SVI, stroke volume index; %T CI ≥ 2.5, percentage of time with CI value ≥ 2.5 L.min^−1^.m^−2^; %TPPV < 13%, percentage of time with PPV < 13%; CV, coefficient of variation. Data are presented as means ± standard deviation for normal data.

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
