# Peer review of "A Pilot Study Assessing a Closed-Loop System for Goal-Directed Fluid Therapy in Abdominal Surgery Patients"

_jpm, 2022, doi:10.3390/jpm12091409_

Round 1

Reviewer 1 Report

I read the paper with interest.

Overall, the paper is interesting, especially as it is published by a team without COI with the technology.

Somme minor comments:

1) abstract: you say : FC 250 ml and then in the text 100 ml. What was the amount of FC delivered by the CL system ?

Discussion: arterial lactate normal range ? Any data ? what is normal range

2) Some improvements in english is needed: 

- discussion: first sentence to rephrase

-  P7 l 214: rephrase : iplement directed goal therapy

- conclusion: rephrase ( main text)

Also, it could have been nice to nuence the resuts of your study citing and discussion the following papers 

1) Kamal Maheshwari et al: Anesthesiology: Assisted Fluid Management Software Guidance for Intraoperative Fluid Administration. In this paper, the software of the AFM is the same of the one used in the closedloop and they showed that: Fluid boluses recommended by the software resulted in desired SV increases more often, and with greater absolute SV increase, than clinician-initiated boluses. Automated assessment of fluid responsiveness may help clinicians optimize intraoperative fluid management during noncardiac surgery

2) Joosten et al: JCMC: PMID: 29779129

3) Joosten et al: PMID: 33951140

In the 3 papers (above), the authors used the same software but as a real time clinical decision support system and demonstrated that it clearly outperformed manual titration of FC. This can be cited and discussed.

Thank you

Reviewer 2 Report

In this study of patients scheduled for elective major abdominal surgery, 80% agreement was found on fluid loading between the CL and the anesthetist, concluding that CL cannot replace the physician but can help decision making. Although the aim of the study includes promising and future-oriented innovations, it contains some methodological deficiencies. In the study, the anesthesiologist only evaluated the appropriateness of the fluid bolus decision of the CL by some of the determining criteria. In line with the patient's current hemodynamic needs, the decision to perform a fluid bolus independent of the CL was not considered. The authors should explain the reason and consequences of this. The patient should clearly explain fluid treatments and blood product applications outside the CL. It should be noted how often the algorithm evaluates the fluid response. Was there any difference between laparoscopy and laparotomy regarding CL's correct decision-making?

Clarifying these issues by the authors will enable the study to reveal more precise results.

TRANSLATE with x English
Arabic Hebrew Polish
Bulgarian Hindi Portuguese
Catalan Hmong Daw Romanian
Chinese Simplified Hungarian Russian
Chinese Traditional Indonesian Slovak
Czech Italian Slovenian
Danish Japanese Spanish
Dutch Klingon Swedish
English Korean Thai
Estonian Latvian Turkish
Finnish Lithuanian Ukrainian
French Malay Urdu
German Maltese Vietnamese
Greek Norwegian Welsh
Haitian Creole Persian  
TRANSLATE with COPY THE URL BELOW Back EMBED THE SNIPPET BELOW IN YOUR SITE Enable collaborative features and customize widget: Bing Webmaster Portal Back
